# Recrystallization of CsPbBr_3_ Nanoparticles in Fluoropolymer Nonwoven Mats for Down- and Up-Conversion of Light

**DOI:** 10.3390/nano11020412

**Published:** 2021-02-05

**Authors:** Vladimir Neplokh, Daria I. Markina, Maria Baeva, Anton M. Pavlov, Demid A. Kirilenko, Ivan S. Mukhin, Anatoly P. Pushkarev, Sergey V. Makarov, Alexey A. Serdobintsev

**Affiliations:** 1Department of Physics, Alferov University, Khlopina 8/3, 194021 St. Petersburg, Russia; imukhin@yandex.ru; 2Institute of Machine Engineering, Materials and Transport, Peter the Great St. Petersburg Polytechnic University, Polytechnicheskaya 29, 195251 St. Petersburg, Russia; 3Department of Physics and Engineering, ITMO University, Lomonosova 9, 197101 St. Petersburg, Russia; daria.markina@metalab.ifmo.ru (D.I.M.); mgbaeva@itmo.ru (M.B.); anatoly.pushkarev@metalab.ifmo.ru (A.P.P.); s.makarov@metalab.ifmo.ru (S.V.M.); 4Institute of Automation and Control Processes (IACP), Far Eastern Branch of Russian Academy of Sciences, Ulitsa Radio 5, 690041 Vladivostok, Russia; 5Education and Research Institute of Nanostructures and Biosystems, Saratov State University, Astrakhanskaya 83, 410012 Saratov, Russia; pavlovam@sgu.ru (A.M.P.); alexas80@bk.ru (A.A.S.); 6Ioffe Institute, Politekhnicheskaya 29, 194021 St. Petersburg, Russia; zumsisai@gmail.com

**Keywords:** halides perovskites, nanoparticles, electrospun polymers, photoluminescence, upconversion

## Abstract

Inorganic halides perovskite CsPbX_3_ (X = Cl, Br, and I or mixed halide systems Cl/Br and Br/I) nanoparticles are efficient light-conversion objects that have attracted significant attention due to their broadband tunability over the entire visible spectral range of 410–700 nm and high quantum yield of up to 95%. Here, we demonstrate a new method of recrystallization of CsPbBr_3_ nanoparticles inside the electrospun fluoropolymer fibers. We have synthesized nonwoven tetrafluoroethylene mats embedding CsPbBr_3_ nanoparticles using inexpensive commercial precursors and syringe electrospinning equipment. The fabricated nonwoven mat samples demonstrated both down-conversion of UV light to 506 nm and up-conversion of IR femtosecond laser radiation to 513 nm green photoluminescence characterized by narrow emission line-widths of 35 nm. Nanoparticle formation inside nonwoven fibers was confirmed by TEM imaging and water stability tests controlled by fluorimetry measurements. The combination of enhanced optical properties of CsPbBr_3_ nanoparticles and mechanical stability and environmental robustness of highly deformable nonwoven fluoropolymer mats is appealing for flexible optoelectronic applications, while the industry-friendly fabrication method is attractive for commercial implementations.

## 1. Introduction

Halide perovskites have attractive optical properties such as tunable energy of the radiative recombination in the wide spectral range of 420–824 nm and high quantum yield of photoluminescence (PL) [1]. The nanolasers based on halide perovskites demonstrated the high-quality factor up to 3600 [2] at optical pumping. The perovskite materials were employed for the fabrication of many kinds of optoelectronic devices, including solar cells [3], light-emitting diodes (LED) [4], lasers [5], photodiodes [6], photodetectors [7], etc. All-inorganic cesium lead halide perovskite nanoparticles (NPs) attracted significant attention due to their remarkably high PL quantum yield accompanied by a narrow emission line [8,9,10]. The NP PL peak position depends on both the halogen composition of CsPbX_x_Y_3−x_ and the nanocrystal size [11,12]. The cesium lead halide perovskite NPs can be synthesized by the hot injection method allowing inexpensive and scalable fabrication of monodispersed perovskite NPs colloids [12].

In turn, the hybrid active materials containing NPs should sustain significant mechanical deformations while preserving high optical properties to be used in modern applications, including artificial intelligence [13,14,15], robotics, medical tools and monitors, and devices mounted on irregular surfaces [16,17,18,19]. The materials based on polymers with incorporated NPs satisfying these requirements are promising candidates for the envisioned applications [20]. On the other hand, an electrospinning approach allows the usage of a vast variety of polymers to produce materials that consist of thin (hundreds of nanometers) randomly oriented fibers, which yields a porous fibrous structure with a large surface area [21]. The porous structure is advantageous for wearable applications [22] and can also serve as a matrix for numerous objects, including NPs, both embedded in fibers in a ready state and synthesized in situ upon electrospinning [23], including perovskite NPs [24]. Electrospinning technology is an auspicious tool for the production of composite structures with NPs encapsulated into the polymer fibers that protect the NPs, including perovskites [20,25,26,27,28,29,30,31,32], from the environment. Another great advantage of such material is its flexibility, which can be useful when working with substrates of complex shape or for a rapidly growing field of flexible electronics. High porosity of nonwoven mats allows them to be used as components of wearable electronics thanks to high air permeability.

In this work, we propose a cost-effective and scalable fabrication route of the electrospun fluoropolymer mats containing CsPbBr_3_ NPs. The particular feature of the proposed method is NPs recrystallization during the formation of fluoropolymer material leading to a narrow PL line at 506 nm with the full width at half maximum (FWHM) of 35 nm due to distribution of recrystallized NP size in the range of 5–14 nm. To maximize protection for NPs from atmospheric humidity and enhanced solubility of NPs in nonpolar media, we use tetrafluoroethylene-based polymer, which is known for its hydrophobic properties and high thermal and mechanical stability [33]. Fluorimetry measurements data verify the protection of NPs from water.

## 2. Materials and Methods

### 2.1. Synthesis of Perovskite Nanoparticles

Colloidal synthesis of all-inorganic perovskite NPs based on the hot-injection approach was first reported in [12]. In the current work, we used the same method to obtain monodisperse CsPbBr_3_ NPs. The formation of NPs consisted in controlled precipitation of Cs^+^, Pb2^+^, and Br^−^ ions during the reaction of Cs-oleate with a lead (II) halide (PbBr_2_) in a high boiling solvent under N_2_ atmosphere. We used octadecene heated up to 140–200 °C as a solvent. Cs-oleate was a product of the reaction of the Cs_2_CO_3_ and oleic acid in the presence of octadecene. Solubilization of PbBr_2_ and stabilization of the colloidal NPs were achieved by introducing a 1:1 mixture of oleylamine and oleic acid into the 1-octadecene. The nucleation and growth of NPs were fast processes that evolved within several seconds. The size of CsPbBr_3_ NPs can be tuned in the range of 4–15 nm by the reaction temperature (140–200 °C), higher reaction temperature leads to lesser NP sizes.

The synthesized NPs were transferred from 1-octadecene to toluene by centrifugation [34] due to more convenient application and higher stability of NPs in toluene solution in comparison to less volatile 1-octadecene. The NPs/toluene colloid was dried to achieve stoichiometrically pure CsPbBr_3_ material ready to be introduced into the polymer solution for nonwoven material synthesis.

### 2.2. Synthesis of Nonwoven Mats with Perovskite Nanoparticles

Reference nonwoven mats without NPs were electrospun from 12% (wt) solution of vinylidene fluoride-tetrafluoroethylene copolymer (100–150 KDa, Halopolymer, Russia) in a 1:1 mixture of dimethylformamide (DMF) and butyl acetate (BA). Fluoropolymer (powder) was slowly introduced to the prepared solvent while stirring with a magnetic stirrer (400 rpm at 53 °C for 3 h). In this work, we adopted an intuitive approach that should preserve to some extent the size of recrystallized particles as compared to ones taken for the synthesis. To obtain a solution containing perovskite material, the NPs were first dispersed in BA, then DMF was introduced, and the resulting solution was used to dissolve fluoropolymer similarly to the procedure for the reference mats. A typical procedure involved 12 mL of solvent (BA:DMF 1:1), 3.2 mg of NPs, and 1.5 g of fluoropolymer.

Electrospinning was performed using a conventional laboratory syringe setup with a needle connected to the ground terminal of the power supply (HCP 140-65000, F.u.G. Elektronik GmbH, Sande, Germany). A rectangular (15 × 20 cm^2^) aluminum plate, which served as the opposite electrode, was located at the 20 cm distance from the needle, and a negative voltage of 60 kV was applied to it. Fibers were collected on the coated paper fixed to the metal plate electrode. For the electrospinning process, the solution was pumped through the needle using the syringe pump at a flow of 10 mL/h for 30 min.

### 2.3. Microscopic and Optical Characterization

Morphology of the produced nonwoven electrospun mats was studied with MIRA II LMU (Tescan, Brno, Czech Republic) scanning electron microscope (SEM) at the acceleration voltage of 30 kV. For imaging purposes, samples were fixed on the holder and sputtered with gold.

Nichia 365 nm 3 W LED was used to control optical properties of the studied materials and optical imaging.

The phase content was studied using the powder x-ray diffraction method via measurements of 2D XRD patterns. Data were collected with a Kappa Apex II diffractometer (Bruker AXS) using CuK_α_ (λ = 1.54184 Å) radiation generated by a I_μ_S microfocus X-ray tube. The 2D XRD images were converted to Θ-2Θ scans using Dioptas software [35]. The X-ray powder diffraction patterns were collected from CsPbBr_3_ NPs dispersed in the polymer material, and pure polymer material was used as a reference sample.

Continuous PL from the nonwoven mats with embedded NPs was excited by the 365 nm UV mercury lamp. The signal of down-converted UV to visible light was collected from different areas of the sample, and then the spectra were averaged. The fluorescent images of the samples were obtained using Axio Imager A2m (Carl Zeiss) microscope with 50× objectives (Carl Zeiss EC Epiplan-NEOFLUAR). PL spectra were recorded by using an optical fiber spectrometer (Ocean Optics QE Pro) coupled with the aforementioned microscope in a fluorescent regime. The detection area was a spot of 2 μm diameter. Photoluminescence quantum yield (PLQY) was measured using the Shimadzu RF-6000 Spectro Fluorophotometer integration sphere with an excitation wavelength of 488 nm (xenon lamp). PLQY of rhodamine 6G in ethanol solution was measured as the reference [36,37].

IR to visible light up-conversion was excited by Pharos PH2-SP-20W-2mJ single-unit integrated femtosecond laser system and high power optical parametric amplifier Orpheus-F, the signal was measured with the use of imaging spectrograph Andor Kymera 328i. We used the following laser parameters: Wavelength 900 nm, the repetition rate of laser pulses 100 kHz, and pulse duration 220 fs.

To demonstrate the protection of NPs from water, PL was measured from an electrospun mat with NPs using a Synergy H1 plate reader (BioTek, Winooski, VT, USA). For measurements, a piece of the nonwoven mat was put into a standard 96-well plate cover to occupy the area of several individual wells, thus that the data from several points could be averaged. NPs were excited at 380 nm, while emission was collected in the 410–600 nm range. Two series of measurements were performed on the same sample with a 1-week pause. Firstly, a dry sample was measured, then water was added, and PL was measured at several time points within a 4 h time span. Between series, the sample was kept in a fridge at 2–4 °C thus that it dried prior to the second measurement.

Transmission electron microscopy (TEM) measurements were performed to investigate CsPbBr_3_ NPs inside nonwoven mats using JEOL JEM-2100F microscope, with an operating voltage of 200 kV. Cross-section images of nonwoven fibers with and without perovskite NPs were investigated.

## 3. Results and Discussion

The synthesized CsPbBr_3_ NPs in toluene are presented in Figure 1a. The colloid was a green color transparent liquid demonstrating distinguishable PL under white fluorescent lamp lighting. The material and optical properties of the as-synthesized NPs will be studied elsewhere. The photo of dry NPs after toluene evaporation under 365 nm UV 3 W LED lighting is shown in Figure 1a. The NPs colloid in BA under the fluorescent lamp and UV LED lighting is presented in Figure 1b. The liquid demonstrates bright green PL originating from the dispersed NPs. The NP BA colloid has a yellow color different from the green toluene colloid that can be associated with NPs agglomeration. After the introduction of DMF into the NPs BA colloid and stirring, the liquid became transparent and colorless (Figure 1c), which we attributed to the dissolution of NPs into CsBr and PbBr_2_ precursors [38]. In this work, we chose NPs for further recrystallization in nonwoven fibers, considering that stoichiometric halide salts can trigger recrystallization of perovskite NPs, and the process will be simplified.

The synthesized nonwoven mat samples had an area of 25 cm^2^ and a thickness below 20 µm. The macroscopic morphology of the NPs-containing mats was similar to the reference samples without the NPs. The nonwoven mats with NPs had a slightly yellowish color in comparison to the white reference mats (Figure 1d), which can be associated with light absorption in perovskite material. Photographs of the nonwoven samples in the daylight and under the illumination of UV LED are presented in Figure 1d. The down-conversion PL optical microscopic image of nonwoven mats is presented in Figure 1e, and the up-conversion PL was observable with the naked eye, as shown in Figure 1f. The up-converting area corresponded to the laser dot size of 10 µm diameter.

As one can see from the SEM images presented in Figure 2a–d, the mats consisted of ~200 nm thick fibers. No pronounced microscopic morphology changes were induced by the presence of the perovskite particles in the fibers. The TEM images of nonwoven fibers (Figure 2f) demonstrated inclusions with an average diameter of around 10 nm in comparison with the reference fibers (Figure 2e) without any visible features. The inset in Figure 2e shows a TEM image of dried colloid NPs with an average diameter of about 9.8 nm with a standard deviation of 1.8 nm.

Despite the vanishingly small mass fraction of perovskite NPs in fibers and the strong polymer scattering, the reference-subtracted I (2θ) curve presented in Figure 2f inset allowed identification of the characteristic perovskite diffraction. The Strongest Bragg reflections in the XRD pattern, depicted with gray dash lines, corresponded to the CsPbBr_3_ perovskite material ((100), (200), and (211)) [39] with presumably orthorhombic symmetry, as previously observed for nanocrystalline CsPbBr_3_ [40,41].

The PL spectra of colloid NPs and NPs in nonwoven mats measured under UV illumination are shown in Figure 3a. The homogeneous PL signal from the sample indicated a uniform distribution of NPs in the sample volume. The typical PL peak of mat samples was centered at λ_max_ = 506 nm with the FWHM of 35 nm (Figure 3b). The nonwoven material without NPs showed the absence of photoluminescence. Regular PL spectrum FWHM of CsPbBr_3_ NP array with homogeneous size distribution did not exceed 18 nm [42], which was in perfect agreement with our results of PL measurements of toluene colloid NPs presented in Figure 3a. Therefore, we associated the experimental PL spectra broadening with a set of spectra originating from NPs of different sizes. To derive the NP size distribution, we carried out the fitting of the obtained luminescence line shape using Gauss function describing monodisperse NP array [12], the resulting size distribution is shown in Figure 3c. The positions of the edge PL peaks (483 and 529 nm) gave the NP size distribution range of 5–14 nm [43], the average NP diameter was between 9 and 10 nm in good agreement with the TEM measurements (Figure 2f). It should be mentioned that the PL intensity decreased exponentially with time (Figure 3d,e). It may be explained by photo- and thermal degradation of the material at a high density of 365 nm wavelength pumping. PLQY of the as-synthesized CsPbBr_3_ NPs in toluene solution was estimated as 95.4%. The down-conversion PL signal of the NPs in toluene solution over 30 min was measured to be about 30%, which was three times higher than the decay of NP nonwoven mats, which can be attributed to the stronger material degradation of the toluene solution NPs due to a higher PLQY. We also considered that the decay of PL signal of nonwoven mats can be attributed to CsPbBr_3_ chemical reaction with the fiber polymer, especially under UV illumination.

The NPs are known to demonstrate effective photon energy up-conversion under two-photon excitation [44]. The representative measured PL spectrum is shown in Figure 3f. The corresponding λ_max_ = 513 nm and FWHM = 30 nm of PL spectra are in good agreement with the down-conversion results discussed above. The wavelength redshift of 7 nm compared to down-conversion spectra can be explained by the lesser carrier population of the conductive band states due to a less effective two-photon absorption process compared to above-band-gap UV pumping [44]. The PL signal was reduced while being measured in the transmission configuration due to light scattering in nonwoven material. However, the up-conversion PL was observable with a naked eye, as shown in Figure 1f. The PL peak dependency on the excitation fluence in the log-log scale demonstrated a 1.9 slope close to 2, which was typical for a two-photon process [45,46,47]. The up-conversion PL did not demonstrate degradation in time during the measurement, which can be attributed to the low light-matter interaction intensity of the two-photon process (absorption of two photons is a very rare event compared to UV light absorption).

Further investigations on NPs embedded into a matrix of electrospun fibers stability in the atmosphere and when immersed in deionized water were performed using fluorimetry measurements (stability of toluene CsPbBr3 NPs was studied elsewhere, e.g., in [12]). Two series of measurements on the same sample were performed in the dry and wet states that showed signal decay of up to ~33% (Figure 4). Signal intensity was related to its initial maximum value measured immediately after first immersion in water. Interestingly, upon immersion into water, the PL signal increased up to 44% compared to the dry mats, which can probably be attributed to scattering cancellation, which proposes that up-conversion efficiency can be potentially enhanced by the right choice of compound filling interfiber space.

## 4. Conclusions

We have demonstrated the CsPbBr_3_ nanoparticle recrystallization during electrospinning of fluoropolymer fibers for the first time to our best knowledge. The synthesized 5 × 5 cm^2^ nonwoven mats containing NPs support bright down-conversion of 365 UV light to 506 nm green light. NPs/fluoropolymer mats exhibited observable with the naked eye up-conversion of IR femtosecond laser irradiation to 513 nm green light at moderate IR fluence of 1 mJ/cm^2^. TEM and SEM images, PL spectra do not confirm the presence of micrometer size crystals or agglomerates of perovskite nanoparticles visible on the fiber surface. In accordance with the deconvolution analysis of PL spectra, we conclude the CsPbBr_3_ material is crystallized inside the nonwoven fibers in nanoparticles of 9.5 mean diameter and size distribution range of 5–14 nm, which is proven by TEM imaging. Water stability test demonstrated 40% PL intensity decay over 9 h.

Thus, the composite system studied can be considered an encouraging solution for the implementation of NPs in various applications for optoelectronics (including flexible applications).

## Figures and Tables

**Figure 1 nanomaterials-11-00412-f001:**
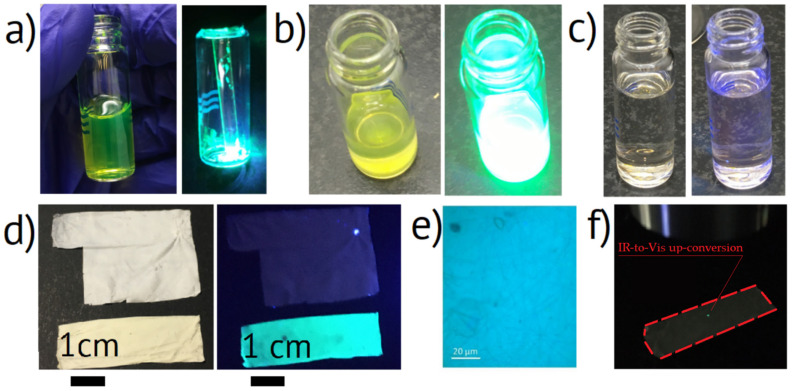
Photographs of (**a**) (left) as-synthesized CsPbBr_3_ NPs transferred to toluene and (right) dry CsPbBr_3_ NPs under UV lighting, (**b**) CsPbBr_3_ NPs in BA under (left) fluorescent and (right) UV lighting, (**c**) CsPbBr_3_ NPs in 1:1 BA/DMF solution under (left) fluorescent and (right) UV lighting, (**d**) nonwoven mats (up) without and (down) with recrystallized CsPbBr_3_ NPs (left) in the daylight and (right) under UV lighting, (**e**) PL image of nonwoven mats, (**f**) IR to visible light up-conversion by CsPbBr_3_ NPs in nonwoven mat, dash line marks the studied sample.

**Figure 2 nanomaterials-11-00412-f002:**
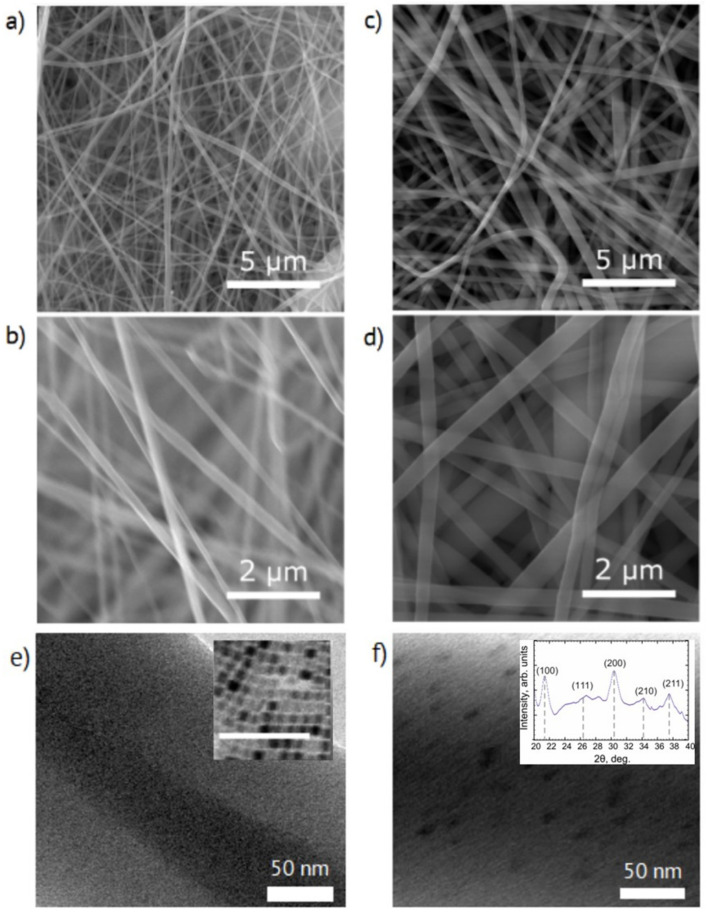
(**a**,**b**) SEM, and (**e**) TEM images of the electrospun mats without perovskite nanoparticles (NPs) and (**c**–**f**) with NPs. Inset in (**e**) demonstrate TEM image of as-synthesized NPs, the scale bar is 100 nm. Inset in (**f**) shows X-ray powder diffraction pattern of CsPbBr_3_ NPs in nonwoven fibers.

**Figure 3 nanomaterials-11-00412-f003:**
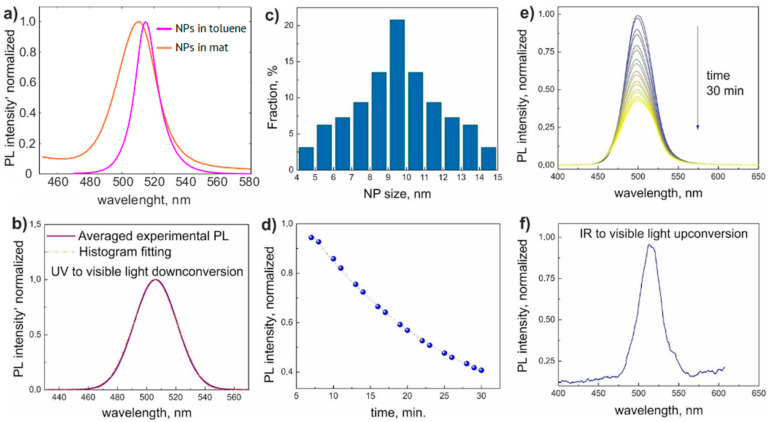
(**a**) Normalized spectra of NPs in a toluene solution (violet line) and in a nonwoven mat (orange line). (**b**) Averaged down-conversion PL intensity and its fitting by signal of monodispersed NPs. (**c**) Calculated distribution for a fraction of different sizes of NPs derived from the monodisperse fitting. (**d**) Time evolution of down-conversion PL spectra. (**e**) Peak intensity decay of down-conversion PL. (**f**) Representative spectrum of up-conversion PL signal.

**Figure 4 nanomaterials-11-00412-f004:**
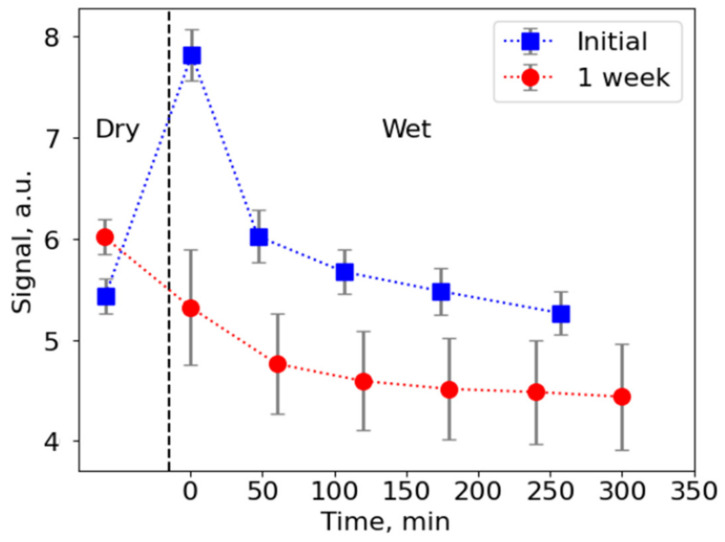
Fluorescence signal from the electrospun mats with NPs, intensity related to the initial maximum value. Two series of measurements on a single sample are shown. Upon initial immersion into water, the PL signal increases up to 44% compared to the dry mats, which can probably be attributed to scattering cancellation.

## Data Availability

The data presented in this study are available on request from the corresponding author. The data are not publicly available due to the author’s readiness to provide it on request.

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
