# Peer review of "Recrystallization of CsPbBr3 Nanoparticles in Fluoropolymer Nonwoven Mats for Down- and Up-Conversion of Light"

_nanomaterials, 2021, doi:10.3390/nano11020412_

Round 1

Reviewer 1 Report

Manuscript Number: nanomaterials-1098522

This work reports on the fabrication route of electrospun fluoropolymer mats containing CsPbBr3 NPs. The samples have characterized by TEM and PL spectra. This work is interesting. I recommend for publication after minor revision.

Comments:

  1. Please check the Figure number in the manuscript. It seems misplaced.  For example, As …..one can see from the SEM images presented in Figure 1 (e), the mats consisted of ~200 nm thick fibers…..
  2. It is recommended to include XRD patterns of respective NPs.

Author Response

Dear Reviewers and Editor,

We thank you for your wise revision, which helped us to greatly improve the presented work and manuscript text. New XRD experiments were performed and the results are added to the text. All the Questions & Answers are discussed below. The revised manuscript is attached in the end of attached 'Answers to Reviewers and MARKED nanomaterials-1098522.pdf' file, the changes are marked with red.

Best regards,

On behalf of the research team,

Dr. Vladimir Neplokh

Reviewer 2 Report

Manuscript ID: nanomaterials-1098522Type of manuscript: ArticleTitle: Recrystallization of CsPbBr3 Nanoparticles in Fluoropolymer Nonwoven Mats for Down- and Up-Conversion of LightAuthors: Vladimir Neplokh *, Daria I. Markina, Maria Baeva, Anton M. Pavlov, Demid A. Kirilenko, Ivan S. Mukhin, Anatoly P. Pushkarev, Sergey V. Makarov, Alexey A. SerdobintsevSubmitted to section: Nanophotonics Materials and Devices  The paper is concerned with a new method of preparation of nano CsPbBr3 particles inside electrospun polymer fibers: recristallization of dissolved CsPbBr3 nanoparticles in the polymer-Cs-Pb-Br precursor solution.The CsPbBr3 nanoparticles in the non woven mats produced show luminescence, up-conversion of IR and down conversion of UV light.  These properties are technologically interesting and the industry friendly new fabrication method is highly attractive. The fibers and theire optical and stability properties are characterized well by photographs under UV and IR illumination, SEM and TEM, PL spectroscopy, and as stability test time dependent PL over 5 hours of fresh samples and again of the same samples after being stored for 1 week. The paper is written well, minor changes are required: 1.  Extend the literature review of research on nanoparticles of CsPbBr3 in fibers in the introduction. A quick look into web-of-science brought up the following citations, that should be checked if appropriate to be mentioned:    

Supersensitive and reusable perovskite nanocomposite fiber paper for time-resolved single-droplet detection

By: Ni, De-Jian; Zhang, Jun; Cao, Zhi-Kai; et al.

JOURNAL OF HAZARDOUS MATERIALS   Volume: ‏ 403     Article Number: 123959   Published: ‏ FEB 5 2021

Full Text from Publisher

Times Cited: 0
(from Web of Science Core Collection)

Usage Count

Show usage counts

Select record2

  1.  

In situ synthesis of coaxial CsPbX3@polymer (X = Cl, Br, I) fibers with significantly enhanced water stability

By: Liu, Wenna; Fu, Hui; Liao, Hao; et al.

JOURNAL OF MATERIALS CHEMISTRY C   Volume: ‏ 8   Issue: ‏ 40   Pages: ‏ 13972-13975   Published: ‏ OCT 28 2020

Full Text from Publisher

Times Cited: 0
(from Web of Science Core Collection)

Usage Count

Show usage counts

Select record3

  1.  

In situ growth of aligned CsPbBr3 nanorods in polymer fibers with tailored aspect ratios

By: Zhang, Huaihao; Fu, Dingfa; Du, Zhentao; et al.

CERAMICS INTERNATIONAL   Volume: ‏ 46   Issue: ‏ 11   Pages: ‏ 18352-18357   Part: ‏ A   Published: ‏ AUG 1 2020

Full Text from Publisher

Times Cited: 1
(from Web of Science Core Collection)

Usage Count

Show usage counts

Select record4

  1.  

Light Down-Converter Based on Luminescent Nanofibers from the Blending of Conjugated Rod-Coil Block Copolymers and Perovskite through Electrospinning

By: Jiang, Dai-Hua; Kobayashi, Saburo; Jao, Chih-Chun; et al.

POLYMERS   Volume: ‏ 12   Issue: ‏ 1     Article Number: 84   Published: ‏ JAN 2020

Free Full Text from Publisher

Times Cited: 3
(from Web of Science Core Collection)

Usage Count

Show usage counts

Select record5

  1.  

Novel ultra-stable and highly luminescent white light-emitting diodes from perovskite quantum dots-Polymer nanofibers through biaxial electrospinning

By: Jiang, Dai-Hua; Tsai, Yi-Hsuan; Veeramuthu, Loganathan; et al.

APL MATERIALS   Volume: ‏ 7   Issue: ‏ 11     Article Number: 111105   Published: ‏ NOV 2019

Free Full Text from Publisher

Times Cited: 7
(from Web of Science Core Collection)

Usage Count

Show usage counts

Select record6

  1.  

Nanoscale optical imaging of perovskite nanocrystals directly embedded in polymer fiber

By: Cha, Ji-Hyun; Kim, Heejin; Lee, Yongjun; et al.

COMPOSITES SCIENCE AND TECHNOLOGY   Volume: ‏ 181     Article Number: 107666   Published: ‏ SEP 8 2019

Full Text from Publisher

Times Cited: 4
(from Web of Science Core Collection)

Usage Count

Show usage counts

Select record7

  1.  

Robust Hydrophobic and Hydrophilic Polymer Fibers Sensitized by Inorganic and Hybrid Lead Halide Perovskite Nanocrystal Emitters

By: Papagiorgis, Paris G.; Manoli, Andreas; Alexiou, Androniki; et al.

FRONTIERS IN CHEMISTRY   Volume: ‏ 7     Article Number: 87   Published: ‏ FEB 26 2019

Free Full Text from Publisher

Times Cited: 5
(from Web of Science Core Collection)

Usage Count

Show usage counts

Select record8

  1.  

A General Strategy for In Situ Growth of All-Inorganic CsPbX3 (X = Br, I, and Cl) Perovskite Nanocrystals in Polymer Fibers toward Significantly Enhanced Water/Thermal Stabilities

By: Liao, Hao; Guo, Shibo; Cao, Sheng; et al.

ADVANCED OPTICAL MATERIALS   Volume: ‏ 6   Issue: ‏ 15     Article Number: 1800346   Published: ‏ AUG 6 2018

Full Text from Publisher

Times Cited: 40
(from Web of Science Core Collection)

Usage Count

Show usage counts

Select record9

  1.  

Perovskite quantum dots encapsulated in electrospun fiber membranes as multifunctional supersensitive sensors for biomolecules, metal ions and pH

By: Wang, Yuanwei; Zhu, Yihua; Huang, Jianfei; et al.

NANOSCALE HORIZONS   Volume: ‏ 2   Issue: ‏ 4   Pages: ‏ 225-232   Published: ‏ JUL 1 2017

Full Text from Publisher

Times Cited: 32
(from Web of Science Core Collection)

Usage Count

Show usage counts

  1. Figure 1: in 1 f) there is not much to see. If the single small bright dot in the center of the grey mat is meant as the one upconverting particle, add an arrow pointing to this particle. If something else is meant, make clear what.
  2. On page 4 last §: Do you really mean (Figure 2f) and (Figure 2e), or should it read (Figure 1f) and (Figure 1e).
  3. Figures 2a),b),c) are never discussed in the text.
  4. Page 5 last line is: the fitting of the obtained curves using Gauss functions

Change to: the fitting of the obtained luminescence line shape using Gauss function

  1. Figure 3b) is never mentioned in the text
  2. Line 196 is: Interestingly, upon immersion into water, the PL signal in-creases up to 44%

7a) change to: Interestingly, upon immersion into water, the PL signal in-creases up to 44% compared to the dry mats.

7b) show non normalized intensities that demonstrate the 44% increase.

  1. Conclusion line 211-212: “no degradation of luminescent properties was observed due to nonwoven mats deformation”.

This is a new result that has not been mentioned nor supported by data in the results part. New results must not appear or be mentioned in the conclusion. Shift to results part and show data, which allow to conclude this statement.

Author Response

Dear Reviewers and Editor,

We thank you for your wise revision, which helped us to greatly improve the presented work and manuscript text. New XRD experiments were performed and the results are added to the text. All the Questions & Answers are discussed below. The revised manuscript is added in the end of attached 'Answers to Reviewers and MARKED nanomaterials-1098522.pdf' file, the changes are marked with red.

Best regards,

On behalf of the research team,

Dr. Vladimir Neplokh

Reviewer 3 Report

This manuscript presents the effect of recrystallization of inorganic halides perovskite nanoparticles for down- and up-conversion of light. The authors showed some experimental results on down- and up-conversion of light from CsPbBr3 nanoparticles, however, there are many missing information. First of all, there are very weak analysis for recrystallization of nanoparticles. In addition, size distribution of perovskite nanoparticles are provided only in calculation. Single value of PLQY (without any supporting information) is insufficient for characterizing the nanoparticles. It is hard to distinguish whether the obtained PL intensity is due to down-conversion or down-shifting.

Author Response

(The authors gave the same response as above.)

Round 2

Reviewer 3 Report

Reviewer's concerns were reflected in the modified manuscript.